# A New Apheresis Device for Antithrombotic Drug Removal during Off-Pump Coronary Artery Bypass Surgery

**DOI:** 10.3390/medicina58101427

**Published:** 2022-10-10

**Authors:** Helmut Mair, Norman Micka, Ferdinand Vogt, Dow Rosenzweig, Frank Vogel, Benedikt Baumer, Stephanie Ulrich, Peter Lamm

**Affiliations:** 1Department of Cardiac Surgery, Artemed Klinikum München Süd, 81379 Munich, Germany; 2Department of Cardiac Surgery, Paracelsus Medical University, 40791 Nuremberg, Germany; 3Department of Anesthesiology, Artemed Klinikum München Süd, 81379 Munich, Germany; 4Department of Cardiology, Benedictus Krankenhaus Tutzing, 82327 Tutzing, Germany

**Keywords:** OPCAB, CytoSorb^®^, ticagrelor, DAPT, adsorption, PUR-01, apheresis pump

## Abstract

*Background and Objectives*: The hemoadsorption device CytoSorb^®^ (CytoSorbents Inc., Princeton, NJ, USA) has been shown to efficiently remove ticagrelor from whole blood in vitro. A promising clinical experience was made with the integration of the hemoadsorption cartridge on the cardiopulmonary bypass (CPB) circuit during cardiac surgery to reduce adverse events. *Materials and Methods*: In this report, we describe a novel approach using a new apheresis platform, PUR-01 (Nikkisio Co., Ltd., Tokyo, Japan), which was used as the extracorporeal circuit where CytoSorb^®^ could be installed for the removal of ticagrelor during off-pump coronary artery bypass (OPCAB) procedures. *Results*: In a 74-year-old male (index case) with coronary artery disease and dual antiplatelet therapy, hemoadsorption was initiated with a skin incision for OPCAB surgery and was continued for 221 min to eliminate ticagrelor. The blood volume that had circulated through the CytoSorb^®^ was 39.04 L in total. Thus far, this treatment strategy has been used in four cases with CHD and DAPT who needed OPCAB surgery. The intraoperative and postoperative courses were uneventful in all patients. No device-related adverse events occurred. *Conclusions*: The combination of the PUR-01 apheresis pump and hemoadsorption with the CytoSorb^®^ column during OPCAB procedures appears to be safe and effective in eliminating antiplatelet drugs.

## 1. Introduction

Increasing numbers of patients with coronary artery disease (CAD) are treated with dual antiplatelet therapy (DAPT) comprising aspirin and a P2Y_12_ inhibitor to reduce adverse events such as myocardial infarctions, ischemic strokes, and cardiovascular death [1,2]. However, patients on DAPT requiring coronary arterial bypass grafting (CABG) have a markedly higher risk of perioperative bleeding and other adverse events [3]. The current guidelines allow for aspirin to be continued through CABG, but require at least 3–7 days of P2Y_12_ inhibitor discontinuation prior to elective surgeries of any type [4], which may not be feasible in urgent or emergency procedures. During this time of waiting, commonly referred to as the “washout” period, intensive inpatient monitoring and the use of “bridging” therapies are usually required to avoid ischemic cardiovascular events.

Within the P2Y_12_ inhibitor class, ticagrelor is the only agent that binds to platelets in a reversible manner. This is an important mechanistic feature with two important implications. First, in cases of bleeding, platelet transfusions are not very useful because ticagrelor may also inhibit the newly transfused platelets. Second, the reversible binding means that ticagrelor may also be available for active removal from the circulation. The hemoadsorption device CytoSorb^®^ (CytoSorbents Inc., Princeton, NJ, USA) has gained considerable attention in the field of cardiovascular surgery as the device has been shown to efficiently remove ticagrelor from whole blood in vitro and early clinical experiences with the integration of the device on the cardiopulmonary bypass (CPB) circuit during cardiac surgery has shown meaningful reductions in adverse events [5,6,7].

Thus far, the clinical use of a hemoadsorption cartridge in cardiac surgery has been limited to on-pump cases with an integration into the CPB circuit. In this report, we describe a novel approach using a new apheresis platform, PUR-01 (Nikkisio Co., Ltd., Tokyo, Japan), which can be used as an extracorporeal circuit where CytoSorb^®^ can be installed for the removal of ticagrelor during off-pump coronary artery bypass (OPCAB) procedures.

## 2. Materials and Methods

The detailed instructions for the use of the PUR-01 apheresis pump are described in *Apheresis Equipment Pure Adjust PUR-01 Operating Instructions* (Manual No. 1055en-R8, 2020-9, Nikkisio Co., Ltd., Tokyo, Japan) [8].

In brief, PUR-01 (Nikkisio Co., Ltd., Tokyo, Japan; Figure 1) is an apheresis machine that has the function of a direct hemoperfusion pump. The machine is loaded with a venous blood tubing line set (ABT-023P Series, Nikkisio Co., Ltd., Tokyo, Japan) and a hemoadsorption device (CytoSorb^®^ 300, CytoSorbents Inc., Princeton, NJ, USA). Figure 2 exemplifies the setup of the system, including the graphical user interface.

The CytoSorb^®^ adsorber consists of a 300 mL cartridge containing biocompatible porous polymer beads that are able to bind hydrophobic substances of a molecular size of up to 60 kDa from whole blood by pore capture and their irreversible surface adsorption properties. The total number of beads in one adsorption cartridge results in a total calculated surface area of more than 40,000 square meters. The adsorber was originally designed to target cytokines and inflammatory mediators [9], but has also been demonstrated to remove antithrombotic medications such as ticagrelor [5] and rivaroxaban [10] as well as bilirubin [11] and myoglobin [12].

A schematic drawing of the extracorporeal circuit is shown in Figure 3: The circuit is equipped with protective systems (e.g., alarms) and features to help ensure patient safety and the correct operation, mainly by monitoring pressure sensors, air, blood, and blood tubing line detectors. The graphical user interface with a touch screen (Figure 1 and Figure 2) guides the user prior to the treatment mode through the installation of the blood lines and the automated priming process using two liters of sodium chloride (NaCl) as a priming solution. Anticoagulation is possible either continuously with a heparin pump or with a single injection of heparin (5000 I.U.) to achieve a recommended clotting time of >180 s. The blood flow rate during treatment can be set from 0 to 200 mL/min. Volume filtration is not possible. During the treatment mode, all important treatment data such as pressure and blood volume are displayed on the screen. In the case of an alarm, a window with a failure indication will pop up.

## 3. Results

### 3.1. Reference Case

In February 2021, a 74-year-old male patient was admitted to our hospital, Artemed Klinikum München Süd, Munich, Germany, for urgent CABG for a severe three-vessel disease (occluded right coronary artery, severe stenosis of the left descending artery, and severe stenosis of the circumflex artery). The patient was treated with DAPT (ticagrelor 2 × 90 mg/day plus aspirin 100 mg/day) for the acute coronary syndrome and non-ST elevation myocardial infarction (NSTEMI) after an acute occlusion of the right coronary artery. The further medical history included hypothyroidism, moderate diverticulitis, hypercholesterolemia, and hypertension. In addition to DAPT, the patient was treated with rosuvastatin, bisoprolol and L-thyroxin. Ticagrelor was stopped the day before surgery.

After the initiation of standard anesthetic care, tubes were connected to a large 12 F, 3-lumen high-flow catheter (Arrow International Inc., Reading, PA, USA), which was implanted into the right cervical vein of the patient for the treatment with the CytoSorb^®^. Adsorption was initiated with a skin incision and was continued for 221 min to eliminate ticagrelor. With the start of PUR-01, a 5000 I.E. single injection of heparin was given. The blood flow rate was set between 150 and 200 mL/h (mean: 176.7 mL/min). Following the sternotomy, left internal thoracic artery (LITA) harvesting was performed. Prior to the bypass anastomosis, another 10,000 units of heparin were administered (activated coagulation time > 300 s). Myocardial revascularization was performed with the OPCAB technique using an Octopus tissue stabilizer (Medtronic, Minneapolis, MN, USA) with the LITA to the left anterior descending artery and venous grafts to the circumflex artery and to the right coronary artery. After the completion of the bypass anastomosis protamine was given for the antagonization of the heparin. The transit-time flow measurement revealed good flow rates of the grafts. The procedure was then finished using standard techniques. The blood volume that had circulated through the CytoSorb^®^ was 39.04 L in total over a treatment duration of 221 min (Figure 4).

The patient was transferred to the intensive care unit and was extubated the same day. Mid-range doses of norepinephrine could be reduced and finally stopped by the end of the first postoperative day. The chest tubes delivered 440 mL in 24 h and were removed on the second postoperative day. Hemoglobin (Hb) dropped from 13.1 g/dL preoperatively to 9.3 g/dL postoperatively. Perioperatively, 2 units of red blood cells were infused. On discharge, the Hb was 12.4 g/dL. Postoperatively, the maximum creatine kinase levels were 232 U/L (normal range, <190 U/L), and the creatine kinase MB (CKMB) isoenzyme was 6.5 µg/L (normal range, <5.2 µg/L). The further postoperative course was uneventful, with a good recovery of the patient. At the 6 weeks follow-up, the patient demonstrated a normal left ventricular function and sinus rhythm, with no cardiac symptoms.

### 3.2. Outcome

Thus far, four patients (all male, age: 61.6 ± 9.4 years) with CHD and DAPT have been operated on with OPCAB (bypass grafts: 2.5 ± 0.3; all patients received the LITA) using a CytoSorb^®^ hemoadsorbtion cartridge integrated into the PUR-01 apheresis device to remove ticagrelor throughout the operation. The blood volume circulating through the CytoSorb^®^ was 28.5 ± 9.0 L over 170 ± 36 min of treatment. Perioperatively, three patients received two units of red blood cells each. One patient needed five units of red blood cells, three units of fresh frozen plasma, and two units of thrombocyte concentrates. This particular patient presented with preoperative anemia, multiple sclerosis, insulin-dependent diabetes, NSTEMI, and a suspected coagulopathy in his medical history.

The intraoperative surgical procedures were not seriously complicated by remarkably enhanced bleeding. No patient had a reoperation for any reason. The postoperative course was uneventful in all patients. No device-related adverse events occurred. The patients were discharged with a renewed prescription for DAPT for the next 6 months. The follow-up time was 9.3 ± 4.8 months.

## 4. Discussion

This is the first report on the intraoperative use of a PUR-01 apheresis pump in combination with a CytoSorb^®^ adsorption column to remove ticagrelor during an OPCAB procedure. The treatment resulted in a good control of the peri- and postoperative bleeding risk and hemodynamic stabilization with a concomitant reduction in norepinephrine requirements as well as an overall satisfactory clinical outcome.

In a previous operation, we had to use a hemodialysis device to act as the extracorporeal circuit due to the lack of an appropriate power unit during an OPCAB operation [7]. The dialysis system alone without a sorbent column does not eliminate ticagrelor as nearly 100% of ticagrelor is protein-bound [5]. With the hemodialysis device, however, only a maximum flow rate of 150 mL/min was possible, probably due to the higher resistance caused by an additional serially configured hemofilter. According to the manufacturer’s recommendations, the 300 mL CytoSorb^®^ column is approved for flow rates between 100–700 mL/min. To avoid any clotting of the column, the CytoSorb^®^ device should be circulated at a minimum flow rate of 100 mL/min. At the maximum possible flow rate, the hemolysis device often set off the alarm and stopped the therapy several times. This was not the case with the PUR-01 machine. A maximum flow rate of 200 mL/min (mean: 175 mL/min) was possible without an interruption to the therapy throughout the whole OPCAB procedure. Various large-lumen tubing sets commercially available for PUR-01 allow even higher blood flow rates.

In OPCAB procedures, the flow rate of PUR-01 occasionally had to be temporarily reduced when intraoperative fluid shifts occurred. During such short periods, norepinephrine was transiently increased and volume was administered, which may have altered the hemoglobin levels postoperatively. At present, there is no evidence regarding the optimal flow rate or blood volume that should pass through the CytoSorb^®^ adsorber for the sufficient adsorption and elimination of ticagrelor from the blood with an extracorporeal circuit. The repeated circulation of blood through the sorbent bead column has been shown to adsorb 99% of the ticagrelor molecules within 3 h in an ex vivo experiment. However, in this study, the velocity through the 300 mL CytoSorb^®^ cartridge was only 17 mL/min [5]. In a clinical study, the adsorber time in a cardiopulmonary bypass was 115 ± 39 min [6]. The blood flow rate was not mentioned in this study; however, in a CPB it is recommended by the company to be at least 400 mL/min. As the adsorber is connected in a parallel circuit to the CPB, CytoSorb^®^ assumes about 12–15% of the blood flow of the CPB. Other clinical trials that have also connected the CytoSorb^®^ column in a parallel configuration to the CPB described a roller pump (actively pumped) controlled flow rate through a 300 mL hemoadsorber of 200 mL/min [13] or an adjustable roller clamp (passive flow) controlled flow rate of 350 mL/min to 600 mL/min through two 300 mL parallel-connected CytoSorb^®^ hemoadsorbers [14] for the elimination of inflammatory substances and markers. However, no real flow rate investigations of the CytoSorb^®^ device have been performed thus far in the field of cardiac surgery. Most of the mentioned flow rates of the CytoSorb^®^ in a passive parallel circuit are determined by the pressure gradient. Recently, Tripathi et al. proved in a benchtop model that, in particular, ticagrelor was effectively removed from whole blood below the therapeutic concentration within 45 min using a flow rate of 300 mL/min [15]. This model was specifically designed to be “life-size” and to mimic the conditions in clinical use as much as possible.

Of note, based on its original purpose, CytoSorb^®^ also offers the possibility to treat hyperinflammation, which inevitably occurs due to the surgical trauma [16]. As the adsorber also eliminates other deleterious substances, it appears plausible that it would be particularly suitable for patients with increased cytokine levels as well as infections and infarctions [17]. PUR-01 also allows the adsorber to be used on patients who do not require a cardiopulmonary bypass, e.g., in visceral, neuro-, or vascular surgery, where the intraoperative adsorption of pathogenic detrimental substances or drugs such as ticagrelor is desirable. Just recently, two reviews [18,19] summarized the current state of hemoadsorption in the cardiovascular field and also discussed the decoupling of the hemoadsorber from the CPB as well as future indications in various medical specialties.

## 5. Conclusions

In conclusion, the combination of the PUR-01 apheresis pump and hemoadsorption with the CytoSorb^®^ column during OPCAB procedures appears to be safe and effective to eliminate antiplatelet drugs and resulted in uneventful intra- and postoperative courses.

## Figures and Tables

**Figure 1 medicina-58-01427-f001:**
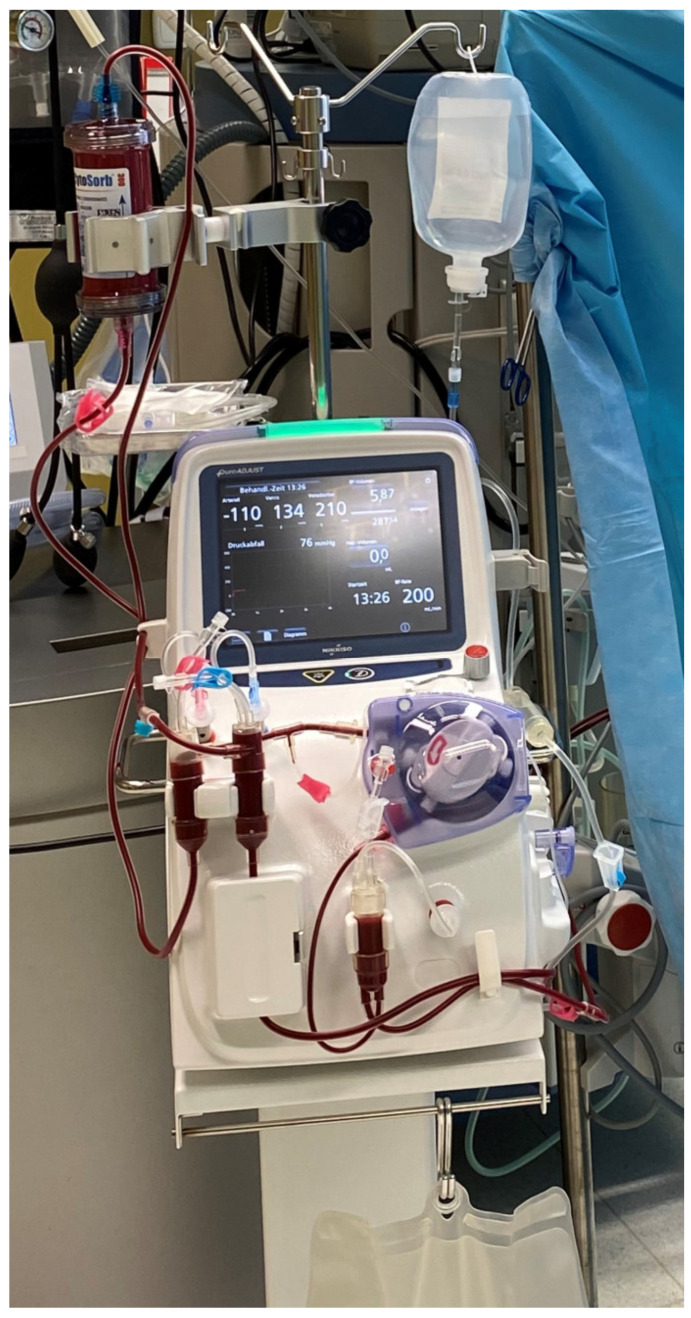
Apheresis machine PUR-01 (Nikkisio Co., Ltd., Tokyo, Japan) loaded with a venous blood tubing line set (ABT-023P Series, Nikkiso Co., Ltd., Tokyo, Japan) and a hemoadsorption device (CytoSorb^®^ 300, CytoSorbents Inc., Princeton, NJ, USA).

**Figure 2 medicina-58-01427-f002:**
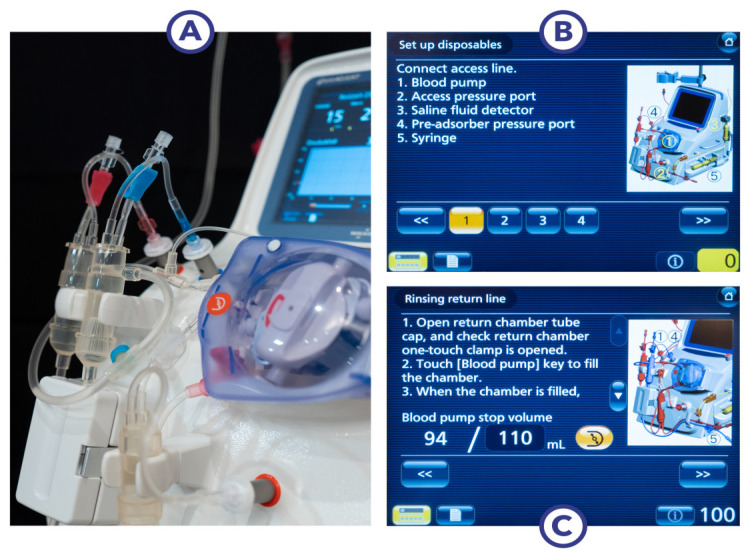
Graphical user interface with touch screen. (**A**) Nikkiso apheresis machine during setup; (**B**) touch screen: setup of the tubing; (**C**) rinsing instructions.

**Figure 3 medicina-58-01427-f003:**
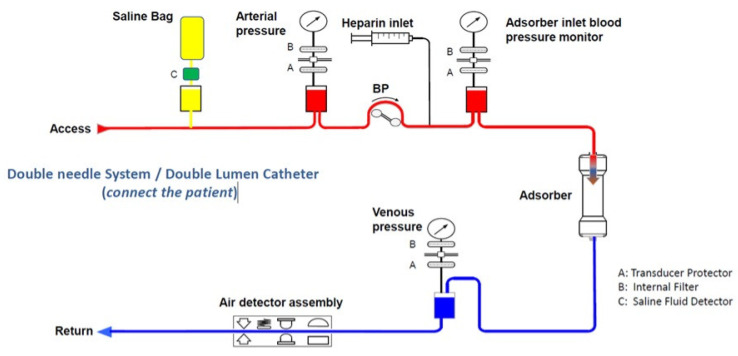
Schematic drawing of the extracorporeal circuit equipped with protective systems and features to help ensure patient safety and correct operation, mainly monitored by pressure (A, B), sensors, air, blood, and blood tubing line detectors. BP: blood pump.

**Figure 4 medicina-58-01427-f004:**
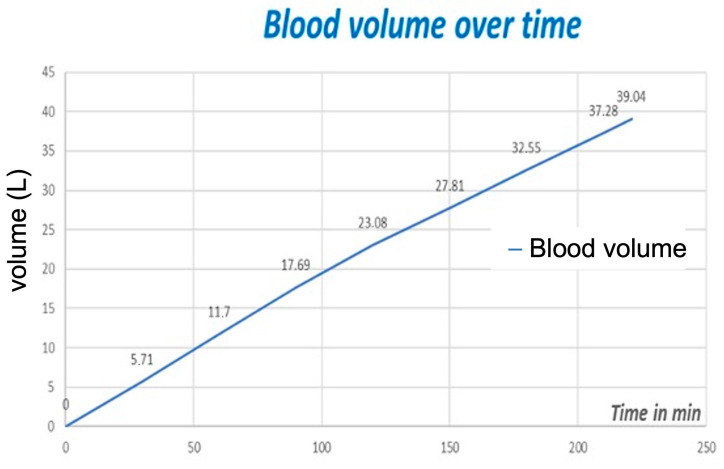
Blood volume processed: in total, 39.04 L over 221 min (treatment duration).

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
