# Peer review of "A New Apheresis Device for Antithrombotic Drug Removal during Off-Pump Coronary Artery Bypass Surgery"

_medicina, 2022, doi:10.3390/medicina58101427_

Round 1
Reviewer 1 Report
The authors present their experience with a novel technique of intraoperative hemoadsorption removing ticagrelor from blood during emergency OPCABG surgery.
The manuscript is well written. However, I do have some minor remarks:
- What´s the meaning of DHP (page5, line 117)? - did the authors mean CPB
- please double-check: the passage outcomes is redundant to the passage in the part before - so please remove 3.2 outcome.
- since how long was the patient on ticagrelor? What was the exact dose? - 90/180mg? For what indication was ticagrelor prescribed and when was the last intake?
- did the aurthors expercienced a clinical, maybe subjective better/shorter intraoperative hemostatis compared to their historical experience with patients under ticagrelor?
Again, well-written and interesting manuscript. I would assume that this could lead to a routine use in emergent/urgent CABG patients under DAPT or ticagrelor treatment undergoing OPCABG surgery.
Author Response
Dear Reviewer, thank you very much for evaluating our paper. Please see the attachment. Sincerely Dr. H. Mair
Reviewer 2 Report
The author has introduced a novel approach for efficient removal of ticagrelor from whole blood using the hemoadsorption device CytoSorb® (CytoSorbents Inc., 12 Princeton, USA). To test the applicability of the device a 74 years old patient was considered for the study. The patient had coronary artery disease. After the whole procedure, no adverse effects were observed and efficient removal of the drug could be done from the blood. Although manuscript is in a good shape and work is of good quality, I suggest the author to add a statement explaining the novelty of the device than the already available devices.
I have a few comments regarding the study:
1. The device has been tested on only one patient; I suggest testing it on some more patients with diverse medical conditions
2. As per the author, to avoid clotting flow rate should be 50 ml/min i.e., similar to a hemodialysis device. How it is better than a hemodialysis device
3. A benchtop model (Tripathi et al., 2022) is already available which is very efficient with a flowrate of 300 ml/min. Explain the novelty of the device developed for the current study.
4. L-66: ‘user interface’ not user’s interface
Author Response
Dear Reviewer, thank you very much for evaluating our paper. Please see the attachments. Sincerely, Dr. H. Mair
